# Simple and Powerful PCG Classification Method Based on Selection and Transfer Learning for Precision Medicine Application

**DOI:** 10.3390/bioengineering10030294

**Published:** 2023-02-26

**Authors:** Ahmed Barnawi, Mehrez Boulares, Rim Somai

**Affiliations:** 1Information Technology Department, Faculty of Computing and Information Technology, King Abdulaziz University, Jeddah 21589, Saudi Arabia; 2Research Laboratory of Technologies of Information and Communication and Electrical Engineering (LaTICE), Higher National School of Engineers of Tunis (ENSIT), University of Tunis, Tunis 1008, Tunisia; 3ESPRIT School of Engineering, Tunis 2035, Tunisia

**Keywords:** CVD classification, data selection, convolutional neural network, pretrained model, deep learning, transfer learning

## Abstract

The World Health Organization (WHO) highlights that cardiovascular diseases (CVDs) are one of the leading causes of death globally, with an estimated rise to over 23.6 million deaths by 2030. This alarming trend can be attributed to our unhealthy lifestyles and lack of attention towards early CVD diagnosis. Traditional cardiac auscultation, where a highly qualified cardiologist listens to the heart sounds, is a crucial diagnostic method, but not always feasible or affordable. Therefore, developing accessible and user-friendly CVD recognition solutions can encourage individuals to integrate regular heart screenings into their routine. Although many automatic CVD screening methods have been proposed, most of them rely on complex prepocessing steps and heart cycle segmentation processes. In this work, we introduce a simple and efficient approach for recognizing normal and abnormal PCG signals using Physionet data. We employ data selection techniques such as kernel density estimation (KDE) for signal duration extraction, signal-to-noise Ratio (SNR), and GMM clustering to improve the performance of 17 pretrained Keras CNN models. Our results indicate that using KDE to select the appropriate signal duration and fine-tuning the VGG19 model results in excellent classification performance with an overall accuracy of 0.97, sensitivity of 0.946, precision of 0.944, and specificity of 0.946.

## 1. Introduction

The World Health Organization (WHO) report [1] states that cardiovascular diseases (CVDs) are a leading cause of death, with 17.3 million deaths annually and an estimate of over 23.6 million deaths by 2030. Early and accurate CVD diagnosis can save lives by reducing the risk of heart failure [2]. One effective method for diagnosing CVDs is acoustic or PhonoCardioGram (PCG) pattern classification. This method recognizes abnormal blood flow sounds from heart valve dysfunction using acoustic signals. However, obtaining accurate results from classical CVD auscultation requires a highly skilled cardiologist. Screenings performed by primary care physicians or medical students have only 40% accuracy [3,4] and even experienced cardiologists have a screening accuracy of only 80% [3,5].

The neglect of regular heart screenings, due to unhealthy lifestyle habits, exacerbates the issue of CVDs. Making accessible and accurate CVD recognition solutions would encourage individuals to integrate regular heart screenings into their daily routine. Many studies have been conducted to diagnose CVDs using PCG signals, with a focus on improving classification results. However, these studies often rely on complex preprocessing steps, optimized heart cycle segmentation, and combined classifier techniques applied to private or modified public PCG datasets. There is no objective comparative benchmark reference for future PCG-based CVD classification.

This paper addresses these issues by presenting a new CVD classification benchmark dedicated to the PCG Physionet dataset and a simple classification architecture based on PCG signal selection with CNN fine-tuning and transfer learning techniques.

The prepocessing of the acoustic signal prior to feeding it into a convolutional neural network (CNN) for classification can significantly impact the accuracy of the results. However, it is important to note that filtering may also remove essential information required by the CNN for proper classification, leading to a reduction in the signal’s dynamic range and obscuring critical spectral features necessary for class differentiation. Our approach leverages strategies that avoid harmful filtering while still improving performance. By carefully selecting the training samples based on sample length and/or signal-to-noise ratio in the prepocessing phase, we have demonstrated the ability to significantly enhance the accuracy of the classification results.

The paper is organized as follows. In Section 2, we present some related work. In Section 3, we introduce the dataset setting and the different data selection methods. In Section 4, we present our classification model. In Section 5, experimental results are presented. In Section 6, we conclude the paper and indicate future and related research directions.

### Contributions

Our research focuses on the classification of normal and abnormal PhonoCardioGram (PCG) signals from the Physionet dataset using Convolutional Neural Network (CNN) technology. Our work presents two main contributions:Development of a common benchmark for Physionet PCG dataset based on CNN transfer learning and fine-tuning techniques. This includes the presentation of classification results such as accuracy, sensitivity, specificity, and precision based on raw Physionet data.Proposal of a simple and effective classification architecture without any prepocessing steps. Our approach is based on a simple PCG data selection technique to improve the normal and abnormal Physionet signal classification results using CNN technology.

## 2. Related Works

Automatic classification of Cardiovascular Diseases (CVDs) is considered a challenging task due to the difficulty in acquiring a large labeled PCG dataset that covers the majority of CVDs. Despite these difficulties, numerous studies have been conducted in recent years. One such study by Grzegorczyk et al. [6] used a hidden Markov model for automatic PCG segmentation and neural networks for PCG signal training. The authors tested their approach on the Physionet dataset [7] and applied pretreatment to eliminate abnormal PCG records. They achieved a classification result with a specificity of 0.76 and a sensitivity of 0.81.

The study by Nouraei et al. in [8] examined the effect of unsupervised clustering strategies, including hierarchical clustering, K-prototype, and partitioning around medoids (PAM), on identifying distinct clusters in patients with Heart failure with preserved ejection (HFpEF) using a mixed dataset of patients. Through the examination of subsets of patients with HFpEF with different long-term outcomes or mortality, they were able to obtain six distinct results.

In [9], the authors conducted a comprehensive review of the relationship between artificial intelligence and COVID-19, citing various COVID-19 detection methods, diagnostic technologies, and surveillance approaches such as fractional multichannel exponent moments (FrMEMs) to extract features from X-ray images [10] and potential neutralizing antibodies discovered for the COVID-19 virus [11]. They also discussed the use of multilayer perceptron, linear regression, and vector autoregression to understand the spread of the virus across the country [12].

Similarly, Chintalapudi et al. in [13] investigated the importance of utilizing machine learning techniques such as cascaded neural network models, recurrent neural networks (RNN), multilayer perception (MLP), and long short-term memory (LSTM) in the correct diagnosis of Parkinson’s disease (PD).

We can also cite the work of [14] who proposed a public challenge based on the Physionet PCG dataset to improve the recognition score, which was initially 0.71 (sensitivity = 0.65, specificity = 0.76). During the competition, 48 teams submitted 348 open source entries and the highest score achieved was 0.86 (sensitivity = 0.94, specificity = 0.78). In the work of [15], the authors proposed a CVD classification technique using the Physionet dataset, which consisted of only 400 heart sound recordings. They relied on the time and frequency domain transformation of the phonocardiogram signal and used a logistic regression hidden semi-Markov model for PCG segmentation. For the classification task, they used and compared three different classifiers: support vector machines, convolutional neural network, and random forest.

In the study of [16], the authors proposed a classification method for cardiovascular diseases (CVD) using deep convolutional neural networks (CNNs) and time/frequency representations of the signals. In the work of [17], the authors used AdaBoost and CNNs to classify normal and abnormal PCG signals from the Physionet dataset. They achieved a sensitivity, specificity, and overall score of 0.9424, 0.7781, and 0.8602 respectively. In [18], the authors proposed a CVD classification based on preprocessing, feature extraction, and training with the Physionet dataset. They used neural networks to classify normal and abnormal signals and obtained a sensitivity of 0.812 and a specificity of 0.860 with an overall accuracy of 0.836.

The study in [19] used the Physionet dataset to perform anomaly detection using signal-to-noise ratio (SNR) and 1D Convolutional Neural Networks. In [20], the researchers presented a heart sound classification technique using multidomain features instead of heartbeat segmentation. They achieved an accuracy of 92.47% with improved sensitivity of 94.08% and specificity of 91.95%. The researchers in [20] used a Butterworth bandpass filter and a pretrained CNN model for CVD classification. In [21], the authors used deep neural network architectures and one-dimensional convolutional neural networks (1D-CNN) with a feed-forward neural network (F-NN) to classify normal and abnormal PCG signals from the Physionet dataset.

In the work of [22], the authors used Logistic Regression-Hsmm for PCG segmentation and feature extraction for CVD classification of normal and abnormal PCG signals from the Physionet dataset. They obtained an accuracy of 79%. In the study of [23], the authors used a pretrained CNN model (AlexNet) and achieved 87% recognition accuracy. The study in [24] aimed to use a nonlinear autoregressive network of exogenous inputs (NARX) for normal/abnormal classification of PCG signals from Physionet. In [25], the authors proposed a deep CNNs framework for heart acoustic classification using short segments of individual heartbeats. They used a 1D-CNN to learn features from raw heartbeats and a 2D-CNN to take inputs from two-dimensional time-frequency features.

## 3. Dataset

In this section, two different PCG datasets are presented. First, the raw Physionet dataset without any data selection process is described. Then, three different data selection methods applied on the original dataset are presented. The goal is to experiment with the impact of selection on the classification results.

### 3.1. Raw Dataset

The publicly available Physionet dataset [14] is a not balanced PCG dataset which contains 665 normal sample and 2575 abnormal sample in WAV format. As shown in Figure 1, the majority of PCG samples are concentrated in the duration range between 8 and 40 s for normal and abnormal class.

If we look at Figure 2, we can deduce that for abnormal class, the highest density of PCG samples is defined at duration 35 s. Concerning the normal class, we can also deduce that the largest concentration of PCG samples are in signal duration 20 s.

Concerning the signal-to-noise ratio (SNR) sample distribution in the function of density (as seen in Figure 3), we can deduce that the highest KDE value of SNR for normal and abnormal classes is zero. This means that the majority of Physionet PCG samples are approximately clean with an acceptable noise signal.

In the same manner, if we look at the Figure 4, it is visually clear that the highest concentration of PCG sample distribution related to normal and abnormal classes in function of SNR is approximately zero.

### 3.2. PCG Data Selection

Based on the different results issued in the previous subsection, in this subsection, we present three main data selection process: data selection based on KDE for optimal signal duration determination, data selection based on optimal SNR, and data selection based on clustering. Notice that we will experiment the impact of these three data selection process on the classification results in the experimentation section.

#### 3.2.1. Data Selection Based on Kernel Density Estimation for Optimal Signal Duration Determination

Kernel density estimation (KDE) [26] is a non-parametric method for estimating the probability density function of a random variable. Given a set of points $X_{i}$ with i=1...n in a *d* dimension space $R^{d}$, the kernel multivariate density estimation is obtained with a kernel K(x) and with window width *h* as following:(1)$$\hat{f}\left(x\right)=\frac{1}{nh^{d}}\sum_{i=1}^{n}K\left(\frac{|X_{i}-x|}{h}\right)$$

With K(u): is a kernel function (using a Gaussian kernel (Formula (Equation 2)). The estimator $\hat{f}\left(x\right)$ determines the percentage of observations closest to a given *x*. If there are several observations close to *x* then $\hat{f}\left(x\right)$ widens. Conversely, if there are only a few $X_{i}$ close to *x* then $\hat{f}\left(x\right)$ remains weak. In other words, the *h* parameter of the Equation (Equation 1), determines the degree of smoothing of the KDE function.
(2)$$k\left(u\right)=e^{-\frac{u^{2}}{2\sigma^{2}}}$$

Based on the discovery issued from the KDE curve shown in Figure 2, the idea is to select all the PCG samples for normal classes with signal duration equal to 20 s and 35 s for abnormal class. As seen in Figure 5, after applying this simple selection process, we obtain 238 PCG samples from abnormal class and 1291 PCG samples from normal class. If we look at the Figure 6 and Figure 7, the obtained PCG samples after the KDE duration selection process for normal and abnormal classes have acceptable SNR values with a high SNR concentration, very close to zero.

#### 3.2.2. Data Selection Based on Optimal SNR

Signal-to-noise ratio (SNR) is defined as the ratio of signal power to the background noise power [27]. Based on the analysis of Figure 3 and Figure 4, which show the highest concentration of SNR related to PCG samples for both normal and abnormal classes, we decided to select PCG samples with SNR greater than or equal to zero. As a result of this selection process, we obtained 221 PCG samples for the abnormal class and 822 PCG samples for the normal class, as shown in Figure 8. Additionally, Figure 9, Figure 10 and Figure 11 provide an overview of the PCG sample distribution in terms of duration after the data selection process with SNR greater than or equal to 0, the KDE curve of PCG samples related to normal and abnormal classes in terms of duration after the SNR greater than or equal to zero in the data selection process, and the PCG sample distribution of normal and abnormal classes in terms of SNR greater than or equal to zero.

#### 3.2.3. Data Selection Based on Clustering

In this part, we chose to use biclustering as our data selection process. The main idea behind biclustering data selection is to suppose that the highest dense cluster constitutes our useful PCG data. In other words, we discard the remaining noise cluster and we preserve only the PCG samples belonging to the big cluster.

For this aim, we have chosen the mixture Gaussian model (GMM) [28] which is a parametric unsupervised clustering model. This model is used for data partitioning into several groups according to the probabilities of belonging and association to each Gaussian characteristics. GMM is based on a mixture of Gaussian models relying on learning the laws of probability that generated the observation data $x_{n}$ (see Equation (Equation 3)).
(3)$$\begin{array}{c} f\left(x_{n}|\theta_{k}\right)=\sum_{k=1}^{M}\pi_{k}N\left(x_{n}|\mu_{k},\sigma_{k}^{2}\right) \end{array}$$

$N\left(x_{n}|\mu_{k},\sigma_{k}^{2}\right)=\frac{1}{{\left(2\pi \right)}^{d/2}\sigma^{1/2}}e^{\left(-\frac{1}{2\sigma_{k}^{2}}{\left(x_{n}-\mu_{k}\right)}^{2}\right)}$, $\pi_{k}\in 1..M$ is the probability of belonging to a Gaussian *k*; k∈1..M ), $\mu {}_{k}\in 1..M$ is the set of the *M* Gaussian averages, $\sigma_{k}^{2}\in 1..M$ the set of covariances matrices, and $\theta_{k}=\pi_{k},\mu_{k},\sigma_{k}^{2}$. Similarly, the multidimensional version of the Gaussian is as follows: $N\left(x_{n}|\mu_{k},\Sigma_{k}\right)=\frac{1}{{\left(2\pi \right)}^{d/2}\Sigma^{1/2}}e^{\left(-\frac{1}{2}{\left(x_{n}-\mu_{k}\right)}^{T}-\Sigma_{k}^{-1}\left(x_{n}-\mu_{k}\right)\right)}$. The best-known method for estimating the GMM parameters ($\pi_{k},\mu_{k}$ and $\sigma_{k}^{2}$), is the iterative method of maximum likelihood calculation (expectation-maximization algorithm or EM [29]). The EM algorithm could be defined through 3 steps:-Step 1: Parameter initialization $\theta_{k}:\pi_{k},\mu_{k},\sigma_{k}^{2}$-Step 2: Repeat until convergence

Estimation step: Calculation of conditional probabilities $t_{ik}$ that the sample *i* comes from the Gaussian *k*. $t_{\left(i,k\right)}=\frac{\pi_{k}N\left(x_{i}|\mu_{k},\sigma_{k}^{2}\right)}{\sum_{j=1}^{m}\pi_{k}N\left(x_{i}|\mu_{j},\sigma_{j}^{2}\right)}$ with j∈1,…,m: the set of Gaussians.Maximization step: Update settings $\theta_{k}^{estim}=argmax_{\theta_{k}}\left(\theta_{k},\theta_{k}^{old}\right)$ and $\pi_{k}^{estim}=\frac{1}{n}\sum_{i=1}^{N}t_{i,k}$,
$\sigma_{k}^{2estim}=\frac{\sum_{i=1}^{N}t_{i,k}{\left(x_{i}-\mu_{k}^{estim}\right)}^{2}}{\sum_{i=1}^{N}t_{i,k}}$, $\mu_{k}^{estim}=\frac{\sum_{i=1}^{N}t_{i,k}x_{i}}{\sum_{i=1}^{N}t_{i,k}}$

The time complexity of EM algorithm for GMM parameters estimation [28,29,30,31] is as following: If *X*: is the dataset size, *M*: the Gaussian number, and *D*: the dataset dimension.

EM estimation step O(XMD+XM).

EM maximization step O(2XMD).

As seen in Figure 12, the result of the selection process based on the highest dense cluster issued from GMM biclustering gives us a 334 PCG sample for the abnormal class and a 1626 PCG sample for the abnormal class. The KDE curve in the function of duration and SNR related to normal and abnormal PCG samples is shown, respectively, in Figure 13 and Figure 14. Furthermore, Figure 15 gives us an overview of the KDE curve in function of SNR for normal and abnormal PCG classes after the GMM data selection process.

## 4. The Process of Our CNN Benchmark

In this paper, we present a CNN classification system based on transfer learning and fine-tuning. Our system starts with the Physionet dataset, which we use to train the model. Figure 16 shows the architecture of our system, which is built on pretrained CNN models from ImageNet dataset. The first step involves transforming the wav PCG signals into mel spectrogram images using an FFT window of 1024 and a sample rate of 44,100. The second step defines the CNN parameters, including a two-class recognition, an input image size of width = 640 and height = 480, a batch size of 5, 30 epochs, and stochastic gradient descent as the optimizer with a learning rate of 0.0001. In the third step, we fine-tune the layers by using convolutional layers from the pretrained CNN models as feature extraction layers. Additionally, we add six layers including a GlobalAveragePooling2D layer for averaging and better representation of our training vector, three dense layers for the full connected network, a BatchNormalization layer to limit covariate shift, and a dense layer with a sigmoid activation function to obtain a classification value between 0 and 1 (probability).

### 4.1. Mel Spectrogram Representation

The fast Fourier transform is a powerful method to decompose acoustic signal amplitude over time into a multifrequency non periodic signal. However, if we need to represent the spectrum of these frequencies in function of time, we need to perform FFT over several windowed partitioned segments of the input signal. In fact, inspired by measured responses from the human auditory system, studies [32,33,34,35] have shown that humans perception does not perceive the frequencies on a linear scale. For this reason, a dedicated unit to transform frequencies was proposed by Stevens, Volkmann, and Newmann in 1937. This is called the mel scale, which performs mathematical operation on frequencies to convert them to mel scale. In order to obtain the mel spectrogram, we perform the following steps (as seen in Figure 17:Specify the signal into short frames.Windowing in order to reduce spectral leakage.Work out the discrete Fourier transformation.Applying filter banks.Applying the log of the spectrogram values to obtain the log filter-bank energies.Applying discrete cosine transform to decorrelate the filter bank coefficients.

In this work, we have chosen MFCC signal by converting the output features into a png image, which will be applied to the CNN classifier. Figure 18 gives an overview of a normal and abnormal MFCC representation of the input PCG signal.

### 4.2. CNN Models

Recently, deep learning and more especially convolutional neural network (CNN) has trended as an image analysis and classification tool. In fact, many research has [36,37,38,39] have been conducted using CNN to propose neural network models that enable powerful image classification results. Moreover, it is known that CNNs can perform high-level feature extraction while tolerating image distortion conditions and illumination changes, and can provide invariance of image translation. For these reasons, we chose to adopt CNN as our PCG image trainer and classifier.

In fact, in 1998 LeCun [40] introduced the first CNN architecture, designed to recognize handwritten characters. Since the last decade, due to their satisfactory results in computer vision tasks such as face detection [41,42,43], handwritten recognition [44,45,46], and image classification [47,48,49], CNNs are the most-used technology for classifying images. However, in order to design new powerful CNN models, CNN requires large training datasets. Thanks to the knowledge-transfer technique also known as transfer learning appellation [50], it becomes possible to take the advantages of the already trained CNN models on ImageNet by applying some modifications called fine-tuning. Therefore, we can customize these pretrained CNN models in order to be trained on a small dataset without a huge drop in the classification results.

In our work, we used several pretrained CNN models to classify normal/abnormal PCG spectrogram images. Based on the small public dataset PhysioNet, we fine-tuned and trained the 17 pretrained Keras CNN models (see Table 1). We preserved the convolutional layers which will be used for feature extraction then the additional layers are added:GlobalAveragePooling2D layer for averaging and better representation of our training vector.Three dense layers to define our full connected network.BatchNormalization layer to limit covariate shift by normalizing the activations of each layer.Dense layer with sigmoid activation function in order to obtain classification values between 0 and 1 (probability).

Keras CNN models are trained on the following dataset using the Google Colab plateform to allow the use of dedicated GPU facilities: 1×Tesla K80, having 2496 CUDA cores, compute 3.7, 12 GB (11.439 GB Usable) GDDR5 VRAM:Raw PhysioNet dataset.PhysioNet dataset with data selection using KDE for duration extraction.PhysioNet dataset with data selection using optimal SNR.PhysioNet dataset with data selection using GMM biclustering.

## 5. Experiments and Results

The effect of selecting data on the accuracy of the classification is being studied. First, we concentrate on training and classifying CNN models using the raw dataset without any data selection. Next, we train our CNN models on the data that has been selected based on a 20 s duration for normal PCG signals and 35 s for abnormal PCG signals. Finally, we examine the impact of selecting data based on SNR greater than 0 in the third section. It is worth mentioning that all the classification results have been obtained by taking the average of the results from the three-fold cross validation.

### 5.1. Classification Using Raw Dataset

After performing CNN training on the raw Physionet dataset, we can notice that VGG19 gives the best classification results with accuracy = 0.854, sensitivity = 0.860, precision = 0.794, and specificity = 0.860 (as seen in Table 2).

In addition, we can see that the classification results related to InceptionResNetV2 are close VGG19 with accuracy = 0.825, sensitivity = 0.807, precision = 0.748, and specificity = 0.807. Similarly, Figure 19 gives an overview of the validation and training curves related to VGG19 and InceptionResNEtV2. If we look at Figure 20, we can see that, if we consider the training step duration, mobileNet is the fastest CNN model and ResNet101 is the lowest CNN model. On the other hand, we can see that despite the number of layer of VGG19 (best accuracy result) which is 26 (as seen in Table 1) compared to deeper architecture (such as DenseNet201 with 201 layers) VGG19 is slower than DenseNet201 and is ranked as the fourth-slowest CNN model in term of training time.

#### Classification Using Kernel Density Estimation as Data Selection
Method for Signal Duration 20 s Normal and 35 s Abnormal

After performing data selection on Physionet through the use of signal duration extraction with 20 s for normal PCG signals and 35 s for abnormal PCG signals, we trained all the 17 pretrained CNN models (see Table 1 and we obtained the classification results presented in Table 3. We can notice that through the use of this simple data selection, we obtained an enhancement of all the classification results compared to those without any data selection. As seen in Table 3, we obtained an improvement of VGG19 accuracy from 0.854 (raw dataset) to 0.970, for sensitivity from 0.860 to 0.946, for precision from 0.794 to 0.944, and for specificity from 0.860 to 0.946. Similarly, Figure 21 gives an overview of the validation and training curves related to VGG19 and VGG16. In addition, as seen in Figure 22, the training phase related to VGG19 becomes faster (fourth position after mobilenet, inceptionV3 and resnet50) than the one without data selection. This means that this data selection method allows us to speed up the training phase related to VGG19. On the other hand, we performed an experimental test in order to argue the choice of 20 s and 35 s signal duration extraction, respectively, for normal and abnormal signals. In this test we chose a random signal duration extraction value equal to 50 s for normal and abnormal signals. The classification results related to this experiment is shown in Table 4. If we compare the classification results presented in Table 3 and Table 4, we can see that for VGG19 (best model), the accuracy decreases from 0.970 to 0.870, sensitivity decreases from 0.946 to 0.851, precision decreases from 0.944 to 0.801, and specificity decreases from 0.946 to 0.851. All these results support the idea behind our duration selection method (explained in data selection based on kernel density estimation for optimal signal duration determination subsection).

### 5.2. Classification Using Data Selection Based on Optimal SNR

The idea behind this data selection method is to select all the PCG signals with a signal-to-noise ratio greater than or equal to 0. In other words, we experiment the impact of selecting signals with SNR ≥ 0 on the classification result without performing any prepocessing steps or denoising methods. After applying this data selection method, we trained all the 17 pretrained CNN models (Figure 23 gives an overview of training and validation curves related to VGG19, VGG16, DenseNet169, and InceptionResNetV2). As seen in Table 5, we obtained very good classification results with VGG19, VGG16, DenseNet169, and InceptionResNetV2. The best result was obtained with VGG19 (accuracy = 0.96, sensitivity = 0.943, precision = 0.94 and specificity = 0.943). This result is very close to the classification result obtained after applying data selection based on signal duration.

In fact, if we look at Figure 24, we notice that the VGG19 training time is at the fifth position compared to the fourth position obtained with VGG19, trained on 20 s and 35 s normal and abnormal PCG signals. In other words, the best results in term of training time and classification results was obtained using VGG19 trained on 20 s and 35 s normal and abnormal PCG signals.

#### 5.2.1. Classification Using Clustering as Data Selection Method

In this subsection, we investigate the impact of selecting training data using unsupervised biclustering. We used GMM biclustering with the hypothesis to consider the cluster with the maximum number of sample as our training data. As shown in Table 6 and in Figure 25, we obtained good classification results compared to results without using any data selection method. However, if we compare with the previous results, we can conclude that the best results are obtained using signal selection, based on duration 20 s for normal and 35 s for abnormal PCG data. In this configuration, VGG16 gives the best classification metrics compared to the remaining 16 CNN models with an acceptable training time (sixth position) as seen in Figure 26.

#### 5.2.2. Synthesis

We have undergone a general comparative study against the state-of-the-art methods, as summarized in Table 7. As seen in this table, Dominguez et al. [60] achieved good classification results (accuracy of 0.97, sensitivity of 0.93, specificity of 0.95) using a complex recognition methodology based on heartbeat segmentation and a modified version of the CNN AlexNet model. Philip et al. [61] obtained the worst classification results in Table 7, and this is due to the elimination of the complex heart-cycle segmentation step. The majority of the research work presented in this table employed complex segmentation steps in their classification approach, and they obtained accuracy varying from 0.80 to 0.97, sensitivity from 0.76 to 0.96, and specificity from 0.72 to 0.95. In this work, our main contribution is to obtain very good classification results using a simple classification approach without any complex preprocessing steps, without any segmentation process, and without the use of any new CNN architecture. As seen in Table 7, compared to the work of Dominguez et al. [60], we have achieved similar results with an accuracy equal to 0.97, a slightly better sensitivity result of 0.946, and a slightly lower specificity result of 0.946.

## 6. Conclusions and Perspectives

In this work, we presented a simple classification architecture based on a data-selection process designed to recognize normal and abnormal Physionet PCG signals. We compared our work with the state-of-the-art approaches and concluded that using a data selection process based on a signal duration of 20 s for normal and 35 s for abnormal PCG signals obtained very good CNN classification results with an overall accuracy equal to 0.97, an overall sensitivity equal to 0.946, an overall precision equal to 0.944, an overall specificity equal to 0.946. This work was tested only on the most-used binary class dataset Physionet, which can be considered as a limiting factor. We plan to test it on other public or private multiclass datasets. In addition, the feature-selection process can be improved through the exploitation of a large set of ML feature extraction/selection methods. Furthermore, we plan to create our own multiclass PCG dataset which will be trained on a new CNN model created especially for PCG spectrogram images. 

## Figures and Tables

**Figure 1 bioengineering-10-00294-f001:**
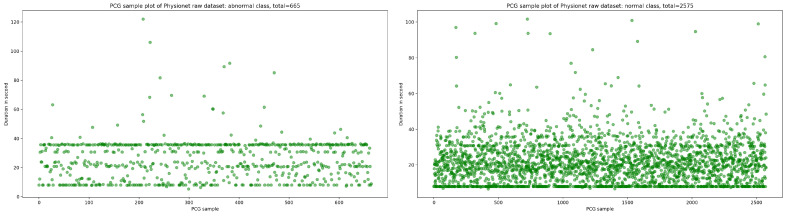
An overview of normal and abnormal sample distribution in function of duration in second.

**Figure 2 bioengineering-10-00294-f002:**
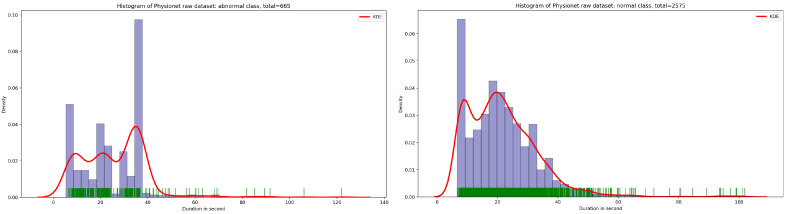
An overview of the kernel density estimation function using Gaussian kernel for normal and abnormal classes.

**Figure 3 bioengineering-10-00294-f003:**
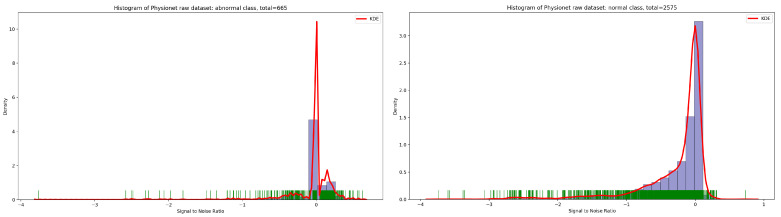
Signal-to-noise ratio in function of density related to normal and abnormal classes.

**Figure 4 bioengineering-10-00294-f004:**
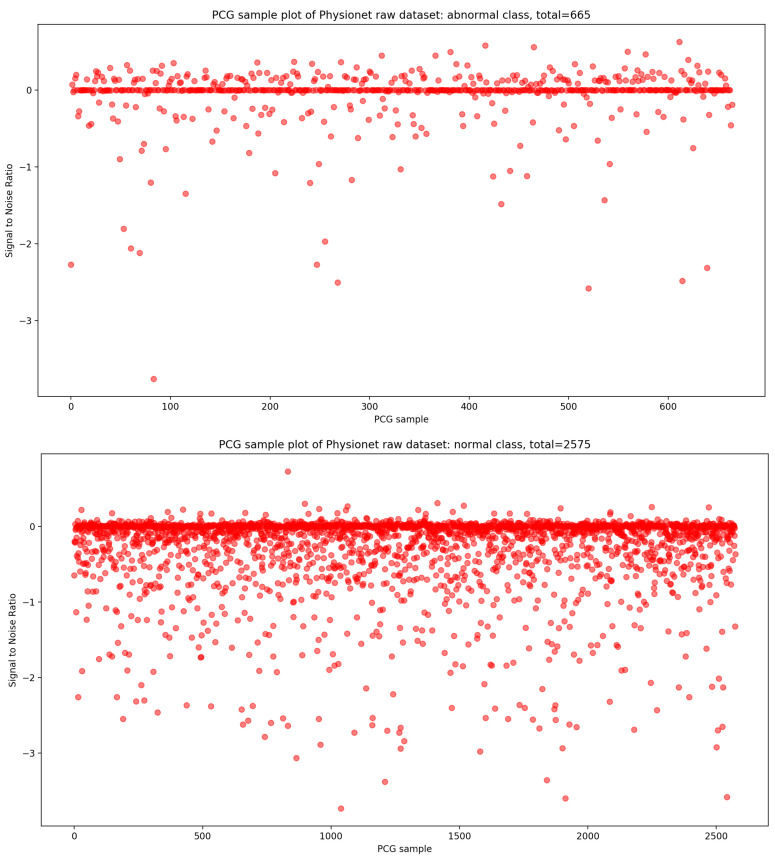
PCG sample distribution in function of signal-to-noise ratio of normal and abnormal classes.

**Figure 5 bioengineering-10-00294-f005:**
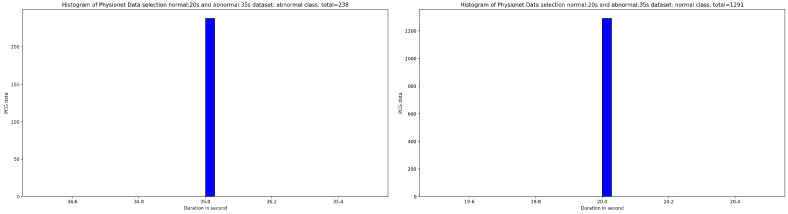
An overview of the PCG sample distribution in function of duration after selecting samples: 35 s from abnormal class and 20 s from normal class.

**Figure 6 bioengineering-10-00294-f006:**
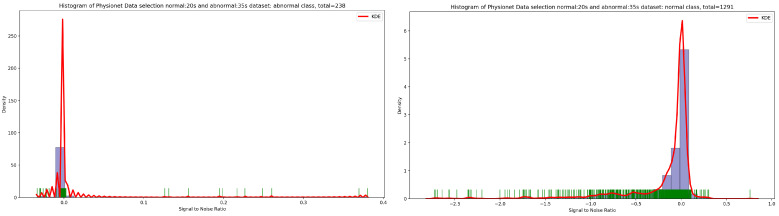
An overview of the SNR distribution in function of KDE density related to normal and abnormal samples after applying the KDE duration selection process.

**Figure 7 bioengineering-10-00294-f007:**
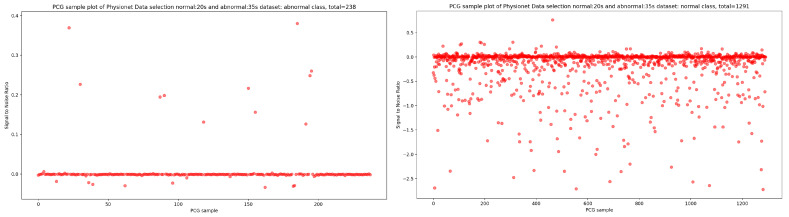
The PCG sample distribution in function of SNR of normal and abnormal classes after KDE duration selection process.

**Figure 8 bioengineering-10-00294-f008:**
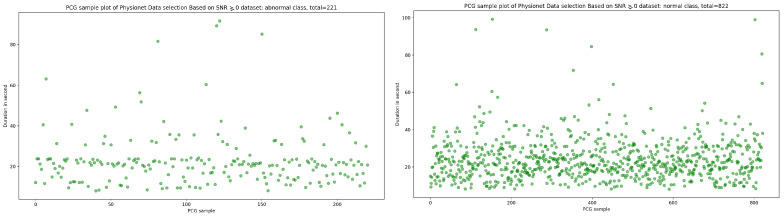
PCG sample distribution in function of duration after SNR greater than or equal to 0 in data selection process.

**Figure 9 bioengineering-10-00294-f009:**
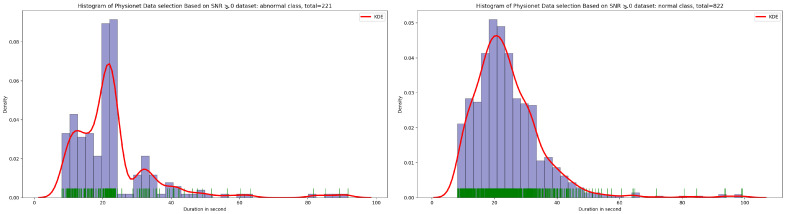
KDE curve of PCG samples related to normal and abnormal classes in function of duration after SNR greater than or equal to 0 in data selection process.

**Figure 10 bioengineering-10-00294-f010:**
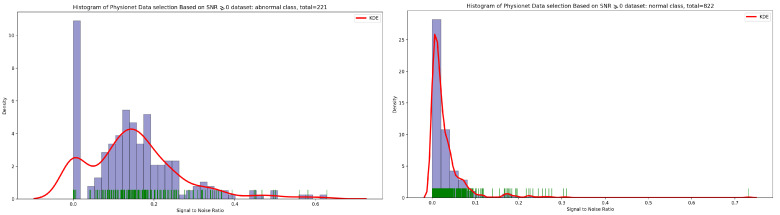
KDE curve of PCG samples related to normal and abnormal classes in function of SNR greater than or equal to 0.

**Figure 11 bioengineering-10-00294-f011:**
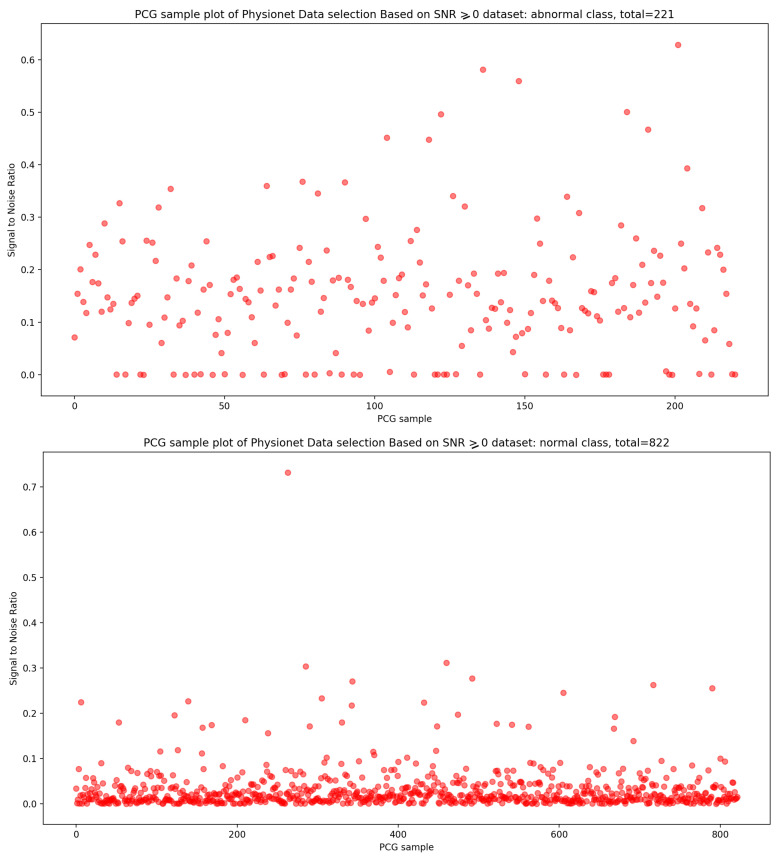
PCG samples distribution of normal and abnormal classes in function of SNR greater than or equal to 0.

**Figure 12 bioengineering-10-00294-f012:**
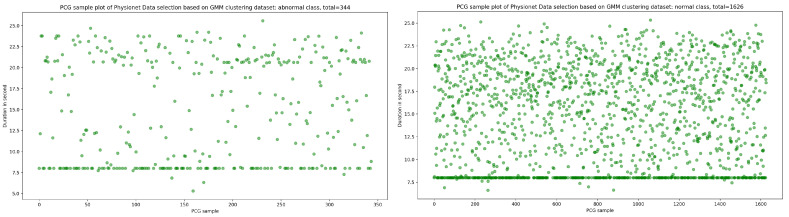
The PCG data distribution of normal and abnormal classes after selecting the highest dense cluster issued from GMM biclustering.

**Figure 13 bioengineering-10-00294-f013:**
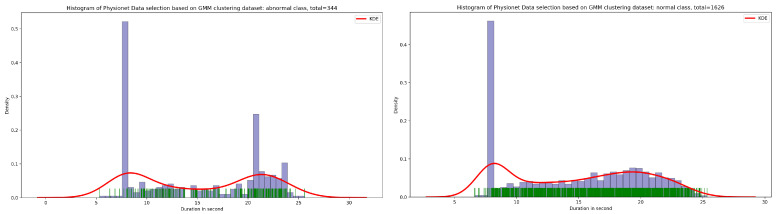
An overview of KDE curve in function of duration for normal and abnormal PCG classes after GMM data selection process.

**Figure 14 bioengineering-10-00294-f014:**
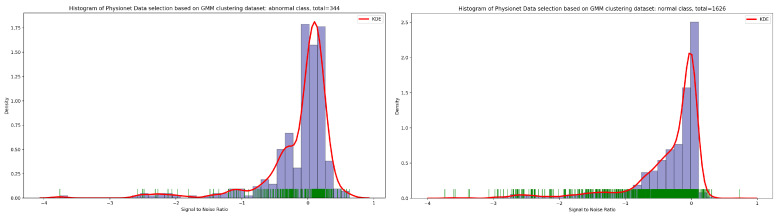
An overview of KDE curve in function of SNR for normal and abnormal PCG classes after GMM data selection process.

**Figure 15 bioengineering-10-00294-f015:**
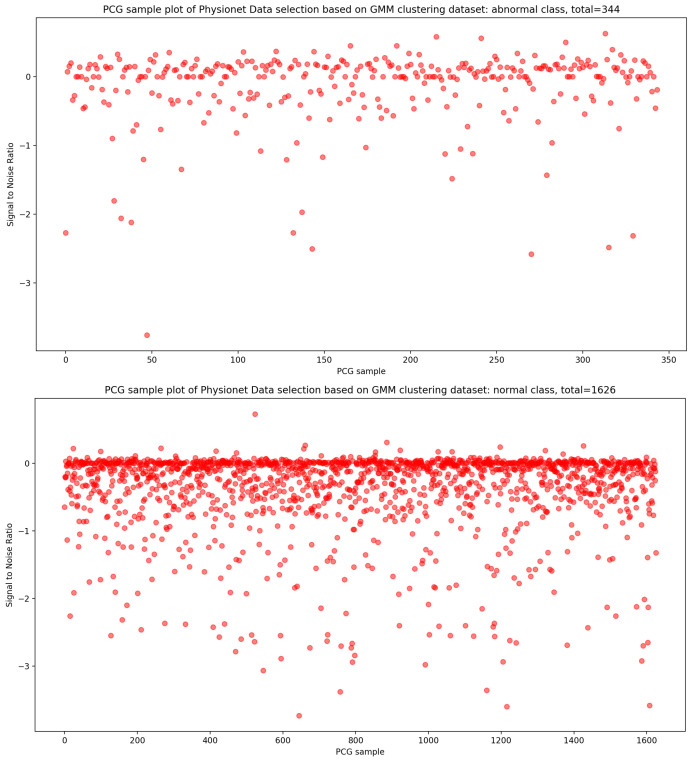
PCG data distribution in function of SNR for normal and abnormal classes after GMM data selection process.

**Figure 16 bioengineering-10-00294-f016:**
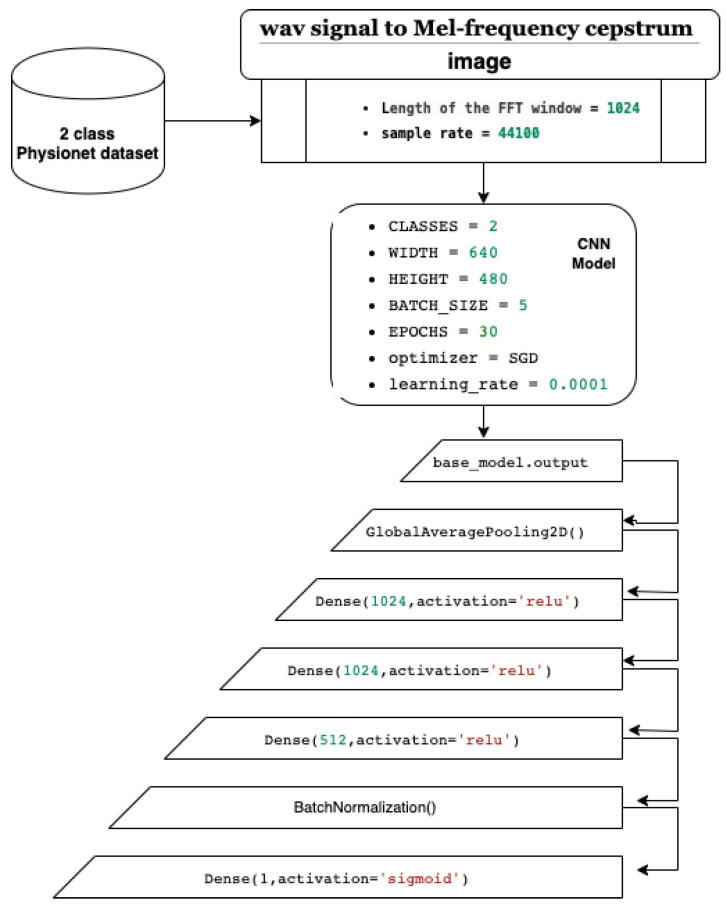
The architecture of our CNN system.

**Figure 17 bioengineering-10-00294-f017:**
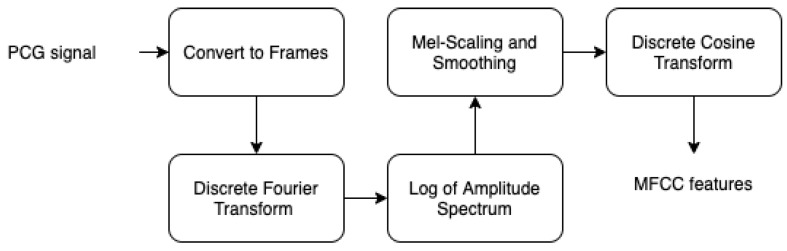
Mel spectrogram steps.

**Figure 18 bioengineering-10-00294-f018:**
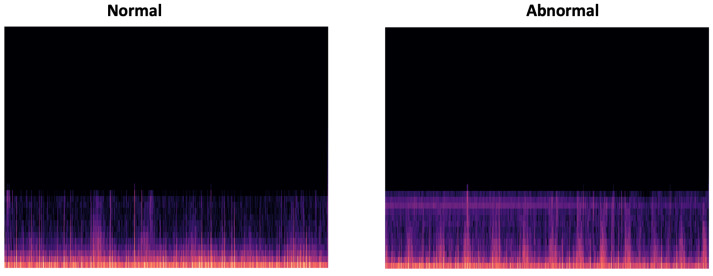
Overview of PCG spectrogram output (normal and abnormal, respectively).

**Figure 19 bioengineering-10-00294-f019:**
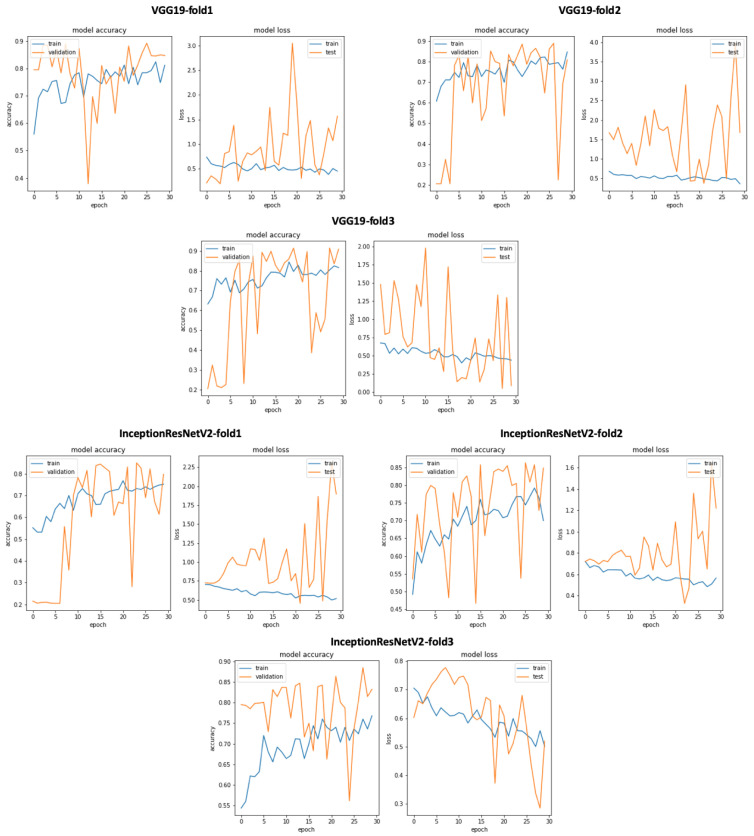
VGG19 and InceptionResNetV2 training and validation curves using raw dataset.

**Figure 20 bioengineering-10-00294-f020:**
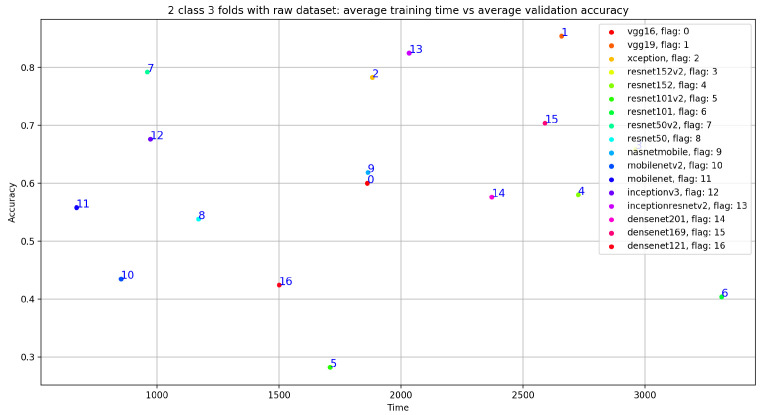
Training time vs. validation accuracy using raw dataset.

**Figure 21 bioengineering-10-00294-f021:**
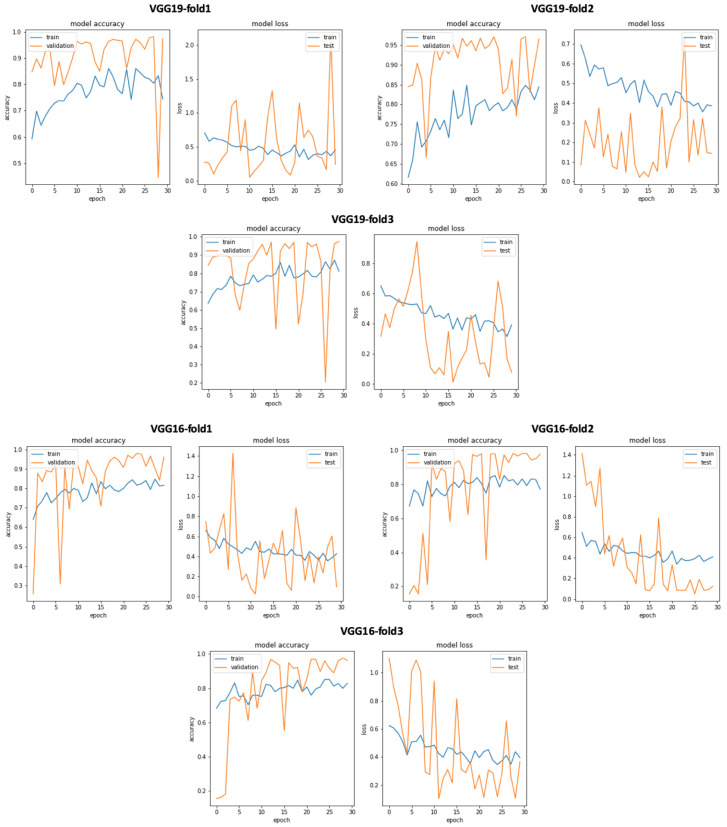
VGG19 and VGG16 training and validation curves using data selection based on KDE.

**Figure 22 bioengineering-10-00294-f022:**
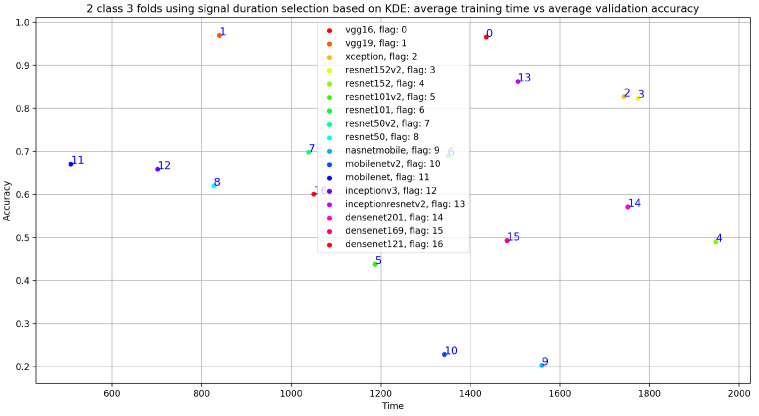
Training time vs. validation accuracy using signal-duration selection based on KDE.

**Figure 23 bioengineering-10-00294-f023:**
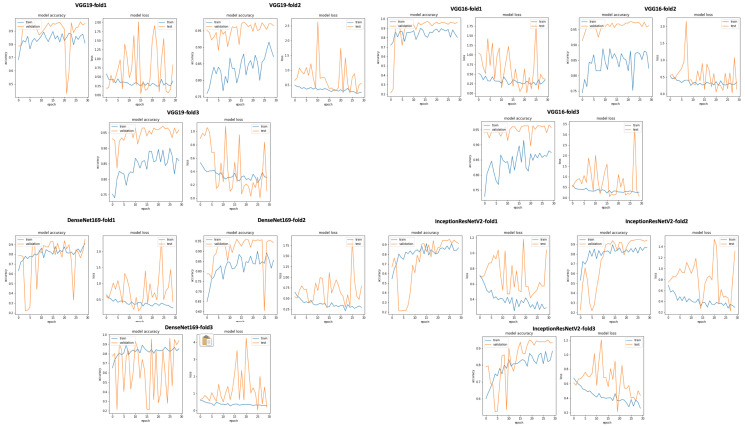
VGG19, VGG16, DenseNet169, and InceptionResNetV2 training and validation curves using data selection based on SNR ≥ 0.

**Figure 24 bioengineering-10-00294-f024:**
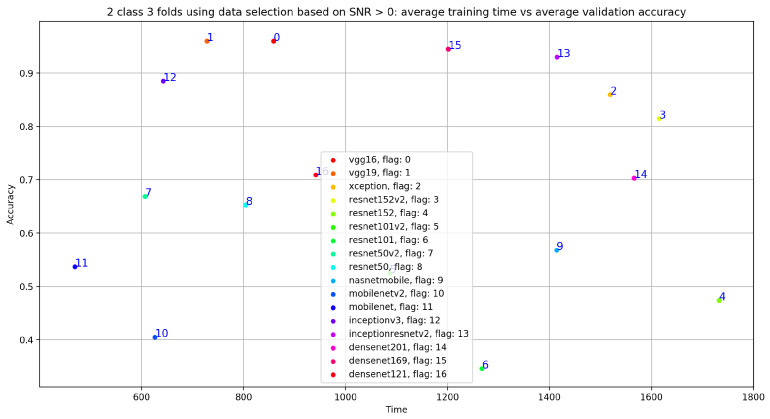
Training time vs. validation accuracy using data selection based on SNR ≥ 0.

**Figure 25 bioengineering-10-00294-f025:**
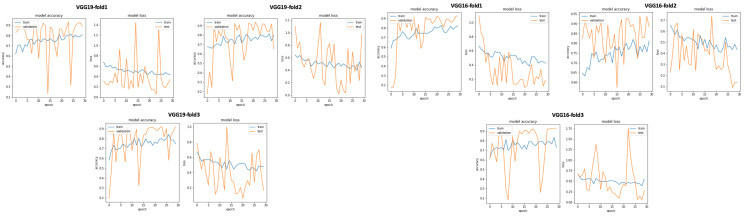
VGG19 abd VGG16 training and validation curves using data selection based on clustering.

**Figure 26 bioengineering-10-00294-f026:**
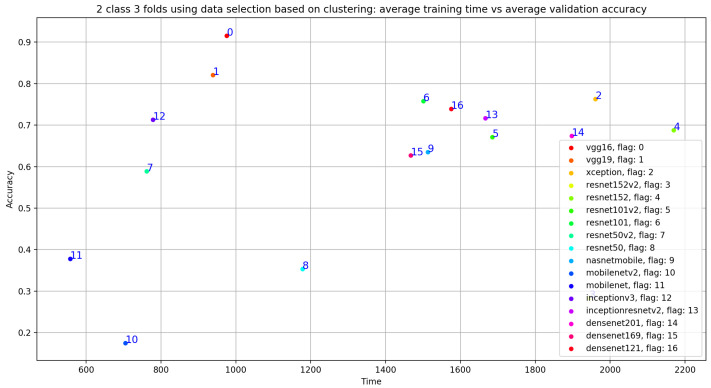
Training time vs. validation accuracy using data selection based on clustering.

**Table 1 bioengineering-10-00294-t001:** Keras CNN models.

Model	Citation	Layers	Size	Parameters
**Xception**	[51]	71	85 MB	44.6 million
**VGG19**	[52]	26	549 MB	143.6 million
**VGG16**	[52]	23	528 MB	138.3 million
**ResNet152V2**	[53]	-	98 MB	25.6 million
**ResNet152**	[53]	-	232 MB	60.4 million
**ResNet101V2**	[53]	-	171 MB	44.6 million
**ResNet101**	[53]	101	167 MB	44.6 million
**ResNet50V2**	[53]		98 MB	25.6 million
**ResNet50**	[53]	-	98 MB	25.6 million
**NASNetMobile**	[54]	-	20 MB	5.3 million
**MobileNetV2**	[55]	53	13 MB	3.5 million
**MobileNet**	[56]	88	16 MB	4.25 million
**InceptionV3**	[57]	48	89 MB	23.9 million
**InceptionResNetV2**	[58]	164	209 MB	55.9 million
**DenseNet201**	[59]	201	77 MB	20 million
**DenseNet169**	[59]	169	57 MB	14.3 million
**DenseNet121**	[59]	121	33 MB	8.06 million

**Table 2 bioengineering-10-00294-t002:** Average metric results related to the raw dataset.

Average	Accuracy	TPR (Sensitivity)	Precision (PPV)	TNR (Specificity)
**VGG16**	0.6	0.527	0.502	0.527
**VGG19**	**0.854**	**0.860**	**0.794**	**0.860**
**Xception**	0.783	0.797	0.714	0.797
**ResNet152V2**	0.659	0.679	0.665	0.679
**ResNet152**	0.580	0.689	0.634	0.689
**ResNet101V2**	0.282	0.537	0.575	0.537
**ResNet101**	0.404	0.585	0.596	0.585
**ResNet50v2**	0.792	0.702	0.702	0.702
**ResNet50**	0.538	0.624	0.638	0.624
**NasNetMobile**	0.619	0.496	0.347	0.496
**MobileNetV2**	0.435	0.476	0.460	0.476
**MobileNet**	0.558	0.595	0.653	0.595
**Inceptionv3**	0.676	0.758	0.673	0.758
**InceptionResNetV2**	0.825	0.807	0.748	0.807
**DenseNet201**	0.576	0.657	0.658	0.657
**DenseNet169**	0.704	0.771	0.715	0.771
**DenseNet121**	0.424	0.622	0.620	0.622

**Table 3 bioengineering-10-00294-t003:** Average metric results related to KDE (duration = 20 s normal, duration = 35 s abnormal) datasets.

Average	Accuracy	TPR (Sensitivity)	Precision (PPV)	TNR (Specificity)
**VGG16**	0.966	0.930	0.946	0.930
**VGG19**	**0.970**	**0.946**	**0.944**	**0.946**
**Xception**	0.828	0.877	0.732	0.877
**ResNet152V2**	0.824	0.873	0.730	0.873
**ResNet152**	0.490	0.667	0.640	0.667
**ResNet101V2**	0.438	0.665	0.422	0.665
**ResNet101**	0.690	0.592	0.812	0.592
**ResNet50v2**	0.698	0.736	0.728	0.736
**ResNet50**	0.620	0.763	0.685	0.763
**NasNetMobile**	0.203	0.489	0.350	0.489
**MobileNetV2**	0.228	0.497	0.526	0.497
**MobileNet**	0.671	0.679	0.673	0.679
**Inceptionv3**	0.659	0.791	0.686	0.791
**InceptionResNetV2**	0.863	0.908	0.765	0.908
**DenseNet201**	0.571	0.725	0.719	0.725
**DenseNet169**	0.493	0.675	0.606	0.675
**DenseNet121**	0.601	0.734	0.714	0.734

**Table 4 bioengineering-10-00294-t004:** Average metric results related to duration = 50 s dataset.

Average	Accuracy	TPR (Sensitivity)	Precision (PPV)	TNR (Specificity)
**VGG16**	0.668	0.747	0.703	0.747
**VGG19**	**0.870**	**0.851**	**0.801**	**0.851**
**Xception**	0.702	0.781	0.689	0.781
**ResNet152V2**	0.501	0.669	0.636	0.669
**ResNet152**	0.785	0.677	0.687	0.677
**ResNet101V2**	0.457	0.628	0.606	0.628
**ResNet101**	0.600	0.616	0.674	0.616
**ResNet50v2**	0.433	0.626	0.611	0.626
**ResNet50**	0.473	0.581	0.636	0.581
**NasNetMobile**	0.451	0.494	0.329	0.494
**MobileNetV2**	0.576	0.535	0.541	0.535
**MobileNet**	0.562	0.680	0.657	0.680
**Inceptionv3**	0.751	0.740	0.729	0.740
**InceptionResNetV2**	0.667	0.687	0.687	0.687
**DenseNet201**	0.694	0.744	0.713	0.744
**DenseNet169**	0.609	0.703	0.699	0.703
**DenseNet121**	0.495	0.637	0.621	0.637

**Table 5 bioengineering-10-00294-t005:** Average metric results related to SNR ≥ 0 dataset.

Average	Accuracy	Sensitivity	Precision	Specificity
**VGG16**	0.960	0.938	0.944	0.938
**VGG19**	**0.960**	**0.943**	**0.940**	**0.943**
**Xception**	0.860	0.895	0.807	0.895
**ResNet152V2**	0.815	0.845	0.790	0.845
**ResNet152**	0.474	0.660	0.665	0.660
**ResNet101V2**	0.525	0.687	0.561	0.687
**ResNet101**	0.346	0.581	0.611	0.581
**ResNet50v2**	0.669	0.773	0.745	0.773
**ResNet50**	0.653	0.746	0.566	0.746
**NasNetMobile**	0.568	0.492	0.344	0.492
**MobileNetV2**	0.405	0.561	0.521	0.561
**MobileNet**	0.537	0.696	0.712	0.696
**Inceptionv3**	0.885	0.893	0.855	0.893
**InceptionResNetV2**	0.930	0.939	0.880	0.939
**DenseNet201**	0.703	0.789	0.612	0.789
**DenseNet169**	0.945	0.938	0.907	0.938
**DenseNet121**	0.709	0.800	0.810	0.800

**Table 6 bioengineering-10-00294-t006:** Average metric results related to clustered dataset.

Average	Accuracy	TPR (Sensitivity)	Precision (PPV)	TNR (Specificity)
**VGG16**	**0.915**	**0.873**	**0.860**	**0.873**
**VGG19**	0.821	0.808	0.787	0.808
**Xception**	0.763	0.795	0.690	0.795
**ResNet152V2**	0.283	0.561	0.590	0.561
**ResNet152**	0.688	0.712	0.728	0.712
**ResNet101V2**	0.671	0.702	0.674	0.702
**ResNet101**	0.758	0.765	0.682	0.765
**ResNet50v2**	0.589	0.666	0.633	0.666
**ResNet50**	0.353	0.576	0.396	0.576
**NasNetMobile**	0.635	0.498	0.561	0.498
**MobileNetV2**	0.175	0.500	0.195	0.500
**MobileNet**	0.378	0.606	0.422	0.606
**Inceptionv3**	0.713	0.773	0.668	0.773
**InceptionResNetV2**	0.717	0.717	0.761	0.717
**DenseNet201**	0.674	0.746	0.672	0.746
**DenseNet169**	0.627	0.758	0.656	0.758
**DenseNet121**	0.739	0.683	0.762	0.683

**Table 7 bioengineering-10-00294-t007:** Comparative analysis of our method with state-of-the-art methods using whole datasets from PhysioNet 2016.

Average	Accuracy	TPR (Sensitivity)	Precision (PPV)	TNR (Specificity)
**our approach**	**0.970**	**0.946**	**0.944**	**0.946**
[62]	0.8697	0.964	-	0.726
[17]	-	0.942	-	0.778
[63]	0.824	-	-	-
[18]	-	0.8095	-	0.839
[16]	-	0.84	-	0.957
[64]	0.852	-	-	-
[65]	-	0.885	-	0.921
[20]	0.879	0.885	-	0.878
[60]	0.97	0.932	-	0.951
[66]	0.915	0.983		0.846
[67]	0.892	0.90	-	0.884
[68]	0.88	0.88	-	0.87
[69]	0.85	0.89	-	0.816
[70]	0.826	0.769	-	0.883
[71]	0.801	0.796	-	0.806
[72]	0.9	0.93	-	0.9
[61]	0.79	0.77	-	0.8

## Data Availability

Not applicable.

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
