# Peer review of "Simple and Powerful PCG Classification Method Based on Selection and Transfer Learning for Precision Medicine Application"

_bioengineering, 2023, doi:10.3390/bioengineering10030294_

Round 1

Reviewer 1 Report

The study proposes a new, simple, and powerful approach based on data selection using signal duration and pre-trained, fine-tuned CNN models to recognize physiologically normal and abnormal PCG signals.
The paper is well written and presented.
I have a few corrections, comments, and suggestions about the paper:
1. Correction: please check this sentence in the introduction: "The screen accuracy results performed by primary care physicians or medical students cannot exceed 40% accuracy [34, 27, ?]." There is a question mark that needs to be removed.
2. There are many previous studies that have been conducted and have similar purposes; to strengthen the contribution of the proposed study, mention its strengths, such as the strength of the method used, and explain the difference compared to the previous studies.
3. Is the only available dataset that can be used to evaluate the proposed model the Physionet dataset? Is there another dataset can be used? If there is another dataset that can be used to evaluate the model, the authors can try to use it, and thus the performance of the proposed model can be found out. Because, when the proposed model is compared to the previous study's models the accuracy achieved is the same with [10] and the sensitivity is still lower than [41] and [15], as well as the sensitivity (lower than [48] and [10]). 
4. In the conclusion, the authors can mention the limitations of the proposed study or model and also mention the challenges for future works.

Author Response

We thank the reviewer for the valuable comments and here are our responses:

[comment 1]: Correction: please check this sentence in the introduction: "The screen accuracy results performed by primary care physicians or medical students cannot exceed 40% accuracy [34, 27, ?]." There is a question mark that needs to be removed.

[response 1]: correction was made

[comment 2]: There are many previous studies that have been conducted and have similar purposes; to strengthen the contribution of the proposed study, mention its strengths, such as the strength of the method used, and explain the difference compared to the previous studies.

[response 2]:

We thank the reviewer for this comment, while in In table 7. We have shown how our approach improves the accuracy compared with state-of-the-art using the standard datasets PhysioNet 2016., we also addressed this point by producing a paragraph in the introduction that highlight the strength of this approach in terms of simplicity.

The pre-processing of the acoustic signal prior to feeding it into a Convolutional Neural Network (CNN) for classification can significantly impact the accuracy of the results. However, it is important to note that filtering may also remove essential information required by the CNN for proper classification, leading to a reduction in the signal's dynamic range and obscuring critical spectral features necessary for class differentiation. Our approach leverages strategies that avoid harmful filtering while still improving performance. By carefully selecting the training samples based on sample length and/or Signal to noise ratio in the pre-processing phase, we have demonstrated the ability to significantly enhance the accuracy of the classification results.

[Comment 3]: Is the only available dataset that can be used to evaluate the proposed model the Physionet dataset? Is there another dataset can be used? If there is another dataset that can be used to evaluate the model, the authors can try to use it, and thus the performance of the proposed model can be found out. Because, when the proposed model is compared to the previous study's models the accuracy achieved is the same with [10] and the sensitivity is still lower than [41] and [15], as well as the sensitivity (lower than [48] and [10]). 

[Response 3]: We thank the reviewer for the comment, In this study, we aimed to demonstrate that by utilizing our approach of simple pre-processing techniques such as optimal feature selection methodology of the dataset, we were able to achieve comparable results when compared to 17 state-of-the-art works that employ more complex approaches. Our experimentation was conducted toward emphasizing this point and for that we thought of using Physionet dataset, which is widely accepted dataset among the research community to the date it is one of few publicly available PCG dataset, which should be considered as a limitation in our domain. Nonetheless the results produced give good insights on the effectiveness of the approach in comparable with more complex approaches.

[comment 4]: In the conclusion, the authors can mention the limitations of the proposed study or model and also mention the challenges for future works.

[Response 4]: We thank the reviewer, The conclusion section has been revised to address the limitations of our work and outline our plans for future research.

Reviewer 2 Report

The present study offers an empirical comparison of the existing, well-known deep-learning, CNN based architectures concerning Cardiovascular Diseases (CVDs), to recognise Physionet normal and abnnormal PhonoCardioGram (PCG) signals.

- Although I find the detailed experimentation useful, the study doesn't refer to the more recent published studies or the studies targeting this particular problem. The most recent ones used in this study are 4 methods from 2020. However, the corresponding methods / architectures are no directly used for comparison, referencing Table 1. Furthermore, adding those results won't be enough anyway as this study does not refer to any work from 2022 and 2023. With a simple Google Scholar search, I found about 300 articles. Although I am not sure whether all directly works on this particular problem here, it still shows that the most recent, state-of-the-art results should be reported beyond Table 1.

Author Response

[Comment 1] Although I find the detailed experimentation useful, the study doesn't refer to the more recent published studies or the studies targeting this particular problem. The most recent ones used in this study are 4 methods from 2020. However, the corresponding methods / architectures are no directly used for comparison, referencing Table 1. Furthermore, adding those results won't be enough anyway as this study does not refer to any work from 2022 and 2023. With a simple Google Scholar search, I found about 300 articles. Although I am not sure whether all directly works on this particular problem here, it still shows that the most recent, state-of-the-art results should be reported beyond Table 1.

[Response 1] The related works section of our paper has been updated with the addition of a paragraphs to provide a more comprehensive and up-to-date overview of relevant research efforts in the field.

“The study by Nouraei et al. in [47] examined the effect of unsupervised clustering strategies, including hierarchical clustering, K-prototype, and partitioning around medoids (PAM), on identifying distinct clusters in patients with Heart failure with preserved ejection (HFpEF) using a mixed dataset of patients. Through the examination of subsets of patients with HFpEF with different long-term outcomes or mortality, they were able to obtain six distinct results.

In [5], the authors conducted a comprehensive review of the relationship between Artificial Intelligence and COVID-19, citing various COVID-19 detection methods, diagnostic technologies, and surveillance approaches such as fractional multichannel exponent moments (FrMEMs) to extract features from X-ray images [13] and potential neutralizing antibodies discovered for the COVID-19 virus [37]. They also discussed the use of multilayer perceptron, linear regression, and
vector autoregression to understand the spread of the virus across the country [58].

Similarly, Chintalapudi et al. in [8] investigated the importance of utilizing machine learning techniques such as cascaded neural network models, Recurrent Neural Networks (RNN), Multilayer Perception (MLP), and Long Short-Term Memory (LSTM) in the correct diagnosis of Parkinson’s disease (PD).”

Round 2

Reviewer 1 Report

The authors have addressed all comments and suggestions.